# Faunal Diet of Adult Cane Toads, *Rhinella marina*, in the Urban Landscape of Southwest Florida

**DOI:** 10.3390/ani13182898

**Published:** 2023-09-13

**Authors:** Melinda J. Schuman, Susan L. Snyder, Copley H. Smoak, Jeffrey R. Schmid

**Affiliations:** Conservancy of Southwest Florida, 1495 Smith Preserve Way, Naples, FL 34102, USA; melindas@conservancy.org (M.J.S.); susanleachsnyder@gmail.com (S.L.S.); omnirodman@gmail.com (C.H.S.)

**Keywords:** dietary analyses, biological invasions, urban ecosystems, taxonomic resolution

## Abstract

**Simple Summary:**

Urban ecosystems provide habitat to many species, including invasive species such as the cane toad (*Rhinella marina*) that is particularly successful in human-altered landscapes. There have been numerous investigations of the cane toad diet, but most of these studies identified prey items at lower taxonomic resolutions (i.e., order or family). We used higher resolution for prey identification and multiple dietary measures of prey consumption to assess the ecological role of cane toads in the urban landscape of southwest Florida. Prey taxa commonly considered urban pests dominated the diet of cane toads inhabiting golf course communities, and there were differences in prey consumed during the wet and dry seasons of this region. We provided a better understanding of the potential relationships between cane toads and their prey in the urban environment.

**Abstract:**

We investigated the diet of cane toads (*Rhinella marina*) inhabiting urbanized areas in southwest Florida to provide high taxonomic resolution of prey items, contrast toad diets between sampling seasons and sexes, and assess this invasive species’ ecological role in the urban landscape. A pest control agency collected cane toads from two golf course communities in Naples, Florida, USA during November–December 2018 (early dry season) and June–July 2019 (early wet season), and faunal stomach contents were quantified from a random subsample of 240 adult toads (30 males and 30 females from each community and season). Yellow-banded millipedes (*Anadenobolus monilicornis*), big-headed ants (*Pheidole* spp.), and hunting billbugs (*Sphenophorus venatus vestitus*) were the most frequently consumed prey items and had the highest total numbers and/or volume with corresponding highest indices of relative importance. There was considerable overlap in the seasonal prey importance values for each golf course community and little if any difference in the importance values between toad sexes in each community. Nonetheless, big-headed ants were the most important prey in both communities during the wet season, while yellow-banded millipedes were the most important dry season prey in one community and hunting billbugs the most important in the other. Despite limited spatiotemporal sampling effort, our results indicated that cane toad was consuming arthropod taxa considered pests in the urban ecosystem. Further studies are needed to investigate the potential effects of human activities and environmental variability on the cane toad diet and to determine whether cane toads act as a biological control for pest populations.

## 1. Introduction

Urban areas are often seen as ecologically inferior to their natural area counterparts; however, in the age of rapid urbanization, these developed areas have nonetheless become complex ecosystems inhabited by many species with their own adaptations to the novel urban conditions [1,2]. An additional contribution has been the dramatic rise of biological invasions via human introductions [3], and many invasive species are highly successful in landscapes altered by urbanization [4,5,6]. The cane toad (*Rhinella marina*) is one such invasive species that has demonstrated an affinity to human habitation [7,8,9,10,11,12].

The cane toad is a large, toxic amphibian with a native range from southern Texas, USA and western Mexico, through Central America, and into central Brazil [8]. This Neotropical species was introduced as biocontrol for agricultural insect pests and has since become established in Australia, Pacific and Caribbean islands, and southern Florida [13]. Interestingly, attempts to introduce cane toads to agricultural areas in Florida during the 1930s and 40s were unsuccessful [14], but urbanization may have been a factor in their becoming established in the state. Toads were accidentally released near the Miami International Airport in the 1950s [15] and began breeding in rockpits created south of the runways, where they purportedly spread through the extensive man-made canal system in Miami [16]. Subsequent importations and human transport have expanded the cane toad range throughout southern and central Florida (Figure 1, [17,18,19]), where its propensity for urbanization is evidenced by observations around homes, residential lawns, golf courses, and other infrastructure [16,20,21,22,23].

Knowledge of a species’ diet is crucial to understanding its interactions with other species and, in some cases, function in the urban ecosystems it may inhabit [24,25]. Dexter [26] provided one of the first and most taxonomically detailed accounts of cane toad food habits, although her identifications focused on prey taxa of economic importance in Puerto Rican agriculture. Wolcott [27] conducted a similarly detailed study in toad fecal pellets collected on a research station’s “well-kept lawn” in what could be considered one of the first characterizations of an urban diet. The prey species and genera identified by these authors appeared to support the use of cane toads as a biological control agent for agricultural insect pests [28]. Numerous subsequent investigations have documented cane toad diet in native and non-native ranges (Appendix A), but most of these studies identified prey items at lower taxonomic resolutions (i.e., order or family). Sex-specific or seasonal foraging patterns have been documented in some studies [29,30,31,32] but not others [16,33], and these discrepancies may be due to limitations in prey identification. Higher taxonomic resolution may explain a higher degree of dietary variation and therefore identify differences that would not be detected at lower levels of prey resolution [34,35,36]. When practical, diet studies should use high-resolution prey identification because the biotic processes within an ecosystem are essentially interactions between species populations [37,38]. 

There have been three diet studies for cane toads in Florida, all of which were conducted in urban ecosystems. Krakauer [16] characterized the frequency of prey items for toads collected from artificial ponds and canals in the southeastern region, while Meshaka and Powell [33] used an index to classify the relative importance of prey for toads captured from a housing development in the south-central region. The objective of both of these studies was to compare the diet of cane toads with that of the native southern toad (*Anaxyrus terrestris)*. Rossi [20] focused on the cane toad’s impact on native vertebrate species, primarily anurans, and provided the frequency of food items consumed in disturbed riparian and edificarian habitats from the west-central region of the state. Beetles (Coleoptera) and ants (Hymenoptera/Formicidae) were the most frequent or relatively important prey for cane toads in Florida, but this low level of taxonomic resolution does not offer much insight into the ecological function of cane toads in these urban ecosystems.

The purpose of our study was to examine the dietary composition of cane toads inhabiting residential golf course communities in southwest Florida. The objectives were to provide fine-scale taxonomic resolution of prey items, contrast cane toad diets between sampling seasons and sexes, and assess this invasive species’ ecological role in the urban landscape.

## 2. Materials and Methods

### 2.1. Study Area

Our study was located in Naples, Collier County on the southwest coast of Florida, USA (Figure 1). Herein, we use the term “urban” as a general reference to the continuum of human-altered landscapes that includes city centers, residential suburbs, and small towns/villages [39]. Southwest Florida has experienced rapid population growth and urbanization during the timeline of cane toad establishment [40,41] and has become a hotspot for records of this invasive species in the state (Figure 1). Over 70 golf course communities are located in the Naples metropolitan area [42], most of which occur in the developed coastal region. Common characteristics of these communities include residences such as condominiums and/or single-family homes; extensive turfgrass coverage (fairways and lawns) with irrigation; ornamental landscaping with mulched beds and typically non-native foliage; permanent water features for aesthetics and stormwater management; and artificial illumination from street and landscape lighting. Our study area included two golf course communities where major roadways, stormwater canals, and other residential developments bounded each community. The golf course communities requested anonymity and, as such, we designated them as “Community A” and “Community B” in our analyses. 

### 2.2. Data Collection

A licensed pest control agency was contracted by the golf course communities to remove cane toads from their respective properties. Toads were collected by hand between 22:00 and 01:00 during November–December 2018 (early dry season) and June–July 2019 (early wet season). Captured toads were kept in ventilated buckets and later euthanized via cooling then freezing [43]. We selected a random subsample of 30 male and 30 female adult toads from pest control collection events (1 night in each community during each season) for a total of 240 toads used in our diet study. Snout-urostyle length (SUL) was measured to the nearest mm for each toad. Toads ≥90 mm SUL are typically considered adults [8], although individuals in the 60–70 cm size class have been identified as mature [44,45]. Sex was determined by visual inspection of skin color/texture [8,46] and confirmed by internal anatomy during dissection. Contents from the stomach were placed in 80% ethanol for storage. Debris such as rocks, grass, and mulch were recorded but not included in diet analyses. Faunal components were examined with a dissecting microscope and identified to the lowest possible taxon. Representative prey taxa were vouchered and microphotographed for identification/verification by entomological experts via BugGuide [47]. Avian prey identification was described in Schuman et al. [48]. After enumeration, sorted prey items were placed in a graduated cylinder and the settled volume was measured to the nearest 0.1 mL [49]. Trace amounts of prey items were assigned a volume of 0.001 mL for analyses.

### 2.3. Data Analyses

For toad sexes, differences in SUL between seasonal collections in the golf course communities were analyzed with Welch’s ANOVA and Games–Howell post hoc tests using the Real Statistics Data Analysis Tools add-in [50] for Microsoft Excel. For each sampling event, differences in SUL by sex were analyzed with Welch’s *t*-test.

Percent frequency of occurrence (%F), number (%N), and volume (%V) of cane toad prey taxa were determined by
$$\%F=\frac{\left(\mathrm{number}\;\mathrm{of}\;\mathrm{samples}\;\mathrm{containing}\;a\;\mathrm{prey}\;\mathrm{taxon}\right)}{\left(\mathrm{total}\;\mathrm{number}\;\mathrm{of}\;\mathrm{samples}\right)}\times 100,$$
$$\%N=\frac{\left(\mathrm{number}\;\mathrm{of}\;\mathrm{individuals}\;\mathrm{for}\;a\;\mathrm{prey}\;\mathrm{taxon}\right)}{\left(\mathrm{total}\;\mathrm{number}\;\mathrm{for}\;\mathrm{all}\;\mathrm{taxa}\right)}\times 100,$$
$$\%V=\frac{\left(\mathrm{volume}\;\mathrm{of}\;a\;\mathrm{prey}\;\mathrm{taxon}\right)}{\left(\mathrm{total}\;\mathrm{volume}\;\mathrm{of}\;\mathrm{all}\;\mathrm{prey}\;\mathrm{taxa}\right)}\times 100.$$

A percent index of relative importance (%IRI; [51]) was calculated for prey taxon *i* with equation
$$\%\mathrm{IRI}=\frac{100\;\left({\%F}_{i}\;\left[{\%N}_{i}+{\%V}_{i}\right]\right)}{\sum_{i=1}^{n}\left({\%F}_{i}\;\left[{\%N}_{i}+{\%V}_{i}\right]\right)}.$$

Compound indices such as IRI reduce potential bias associated with the component diet measures [52,53]. Examples of bias include numerous small food items contributing very little to the volume of the diet, large infrequent food items with high volume contributions, or items consumed by many individuals contributing little to the number or volume in the samples.

Multivariate analyses were performed on cane toad dietary compositions using PRIMER v6 software [54]. Ambiguous prey taxa, those identified at multiple related levels, were resolved by applying the RPMC (remove parent or merge child with parent; [55]) method depending on the abundances or volumes of the corresponding taxa. The number of individuals and volume for the unambiguous prey taxa of each toad were converted to percent composition using the total number and volume of the respective stomach sample. A percent importance value (%IV; [56]) was calculated for the taxa in each sample:$$\%\;\mathrm{IV}=\frac{{\%N}_{i}+{\%V}_{i}}{2}.$$

Bray–Curtis similarity matrices were generated for the untransformed importance value percentages in each golf course community. Two-way crossed analysis of similarity (ANOSIM; 999 permutations) was used to contrast diet composition between sampling seasons, allowing for differences between toad sexes, and between sexes allowing for differences between seasons. The ANOSIM statistic *R* is an absolute measure of the separation between two or more groups, typically ranging between 0 (no separation of groups) and 1 (complete separation; [54]). Unlike the significance level, the *R* statistic is not unduly affected by the group sample sizes (*R* values close to zero can be deemed significant with a large number of replicates or, conversely, few replicates can result in a large *R* value that is not significant; [57]). Two-way similarity percentages routine (SIMPER) was used to identify prey taxa contributing to the dissimilarities for any significant dietary differences in season or sex for a given golf course community.

## 3. Results

Mean body lengths for female toads were significantly different (F = 5.33, *p* = 0.002) among seasonal sampling events with smaller females collected during the early dry season in golf course Community A (Table 1). The mean body length for male toads was smaller than that of females during the respective sampling events, although the relationship was not significantly different for Community A during the early dry season. The two smallest toads (62 and 73 mm SUL) were identified as males and all others were ≥90 mm in body length.

A total of 13,961 prey items in 180 taxonomic categories (5 phyla with at least 15 classes, 32 orders, 59 families, 78 genera, and 60 species) was identified from the stomachs of 239 cane toads (Appendix A). One of the samples contained sapric organic material but no recognizable faunal components. Ants (Formicidae) accounted for 58% of the numeric total of prey but only contributed 4% to the total prey volume. Beetles (Coleoptera) and millipedes (Diplopoda) accounted for 32% and 31%, respectively, of the volumetric total. Beetles contributed 23% to the total number of prey while millipedes only accounted for 7% of the total number. Yellow-banded millipedes (*Anadenobolus monilicornis*), big-headed ants (*Pheidole* spp.), and hunting billbugs (*Sphenophorus venatus vestitus*) were the most frequently consumed prey items and had the highest total numbers and/or volume with corresponding highest indices of relative importance (combined IRI of 74%) for all taxonomic categories.

After resolving ambiguous taxonomic categories, there were 109 unique prey taxa (4 classes, 9 orders, 15 families, 32 genera, and 49 species), and 14 of these unambiguous taxa had an IRI > 1% for at least one of the seasonal cane toad collections in either of the golf course communities (Table 2 for Community A and Table 3 for Community B). Hunting billbugs were the dominant prey item (57% N, 35% V, 68% F, and 68% IRI) in golf course Community A during the early dry season. Shore earwigs (*Labidura riparia*) were also frequently consumed (62% F) during this season, but their lower contributions to prey volume and number resulted in lower relative importance (16% IRI). In contrast, yellow-banded millipedes were the dominant dry season prey (43% N, 65% V, 56% F, and 80% IRI) in golf course Community B with much lower contributions to the diet measures by hunting billbugs and a corresponding low relative importance (7% IRI). Big-headed ants were the most important early wet season prey in both golf course communities (49% IRI), owing to the high frequency of occurrence and the highest contributions to number of prey items but with relatively low-volume contributions. Yellow-banded millipedes in both communities and lepidopterans, primarily caterpillars, in Community B were also frequently consumed during the wet season and both taxa had the highest prey volume contributions but much lower numbers and corresponding relative importance. Additionally, Cuban May beetles (*Phyllophaga bruneri*) and scavenger scarab beetles (*Hybosorus illigeri*) were only consumed during the wet season in both communities.

For golf course Community A, there was a significant difference (Global *R* = 0.25, *p* = 0.001) in prey importance value percentages between sampling seasons. The low *R* value indicated moderate overlap with some separation in prey compositions. The SIMPER routine identified hunting billbugs as contributing the most to the seasonal diet dissimilarity in Community A with a much greater mean importance value percentage for this prey item in the early dry season (Figure 2a). Yellow-banded millipedes had the second highest contribution to the seasonal dissimilarity, but the lack of seasonal difference between the mean importance value percentages indicated high variability in the consumption of this prey item. Shore earwigs also had substantially higher mean importance value percentages during the early dry season, whereas big-headed ants and Florida carpenter ants (*Camponotus floridanus*) had higher mean percentages during the early wet season. Other distinctive prey items only consumed in the wet season included uneven billbugs (*Sphenophorus inaequalis*) and Cuban May beetles. The average similarity value for prey taxa consumed in the early dry season (26.4%) was higher than that of the early wet season (18.9%), but these relatively low similarity values also indicate considerable variability in prey consumed by toads during each season.

There was a significant difference (Global *R* = 0.09, *p* = 0.001) in prey importance value percentages between toad sexes in golf course Community A; however, the very low *R* value indicated strongly overlapping importance value compositions with little difference between sexes. Nonetheless, hunting billbugs were identified by the SIMPER routine as contributing the most to the dietary dissimilarity between toad sexes and males exhibited a greater mean Importance value percentage for this prey item (Figure 2b). Female toads demonstrated higher importance value percentages for yellow-banded millipedes and big-headed ants. Dietary variability for prey importance values within each sex group was also indicated by the relatively low average similarities (24.8% males and 20.5% females).

For golf course Community B, there was a significant difference (Global *R* = 0.16, *p* = 0.001) in prey importance value percentages between sampling seasons and the low *R* value indicated moderate overlap with some differences in prey compositions. The SIMPER routine identified yellow-banded millipedes as contributing the most to the seasonal dissimilarity in Community B and this prey item had a substantially greater mean importance value percentage during the early dry season (Figure 3a). Hunting billbugs, long-flange millipedes (*Asiomorpha coarctata*), and shore earwigs also had higher importance value percentages during the early dry season but contributed less to the seasonal dissimilarity. Big-headed ants and lepidopterans had higher mean importance value percentages during the early wet season. Other prey items with distinctive wet season differences included Cuban May beetles, scavenger scarab beetles, darkling beetles (*Platydema*), and rover ants (*Brachymyrmex*). Average similarities were higher in the early dry season (23.5%) compared to the early wet season (11.5%), but dietary variability was indicated by the relatively low values.

There was a marginally significant difference (Global *R* = 0.03, *p* = 0.048) in prey importance value percentages between toad sexes in golf course Community B. The negligible *R* value indicated dietary compositions were barely separable with little if any differences in prey importance values between sexes. Furthermore, the borderline statistical significance should be interpreted cautiously owing to the relatively high number of replicates/permutations and their influence on significance levels. The SIMPER routine indicated some minor dietary differences by sex (Figure 3b) with females demonstrating higher importance value percentages for yellow-banded millipedes and males with higher values for lepidopterans and hunting billbugs, among others. These analyses may have also been confounded by variability in prey consumption as indicated by the lack of difference between the mean importance value percentages of big-headed ants and the low similarity values (19.5% for females and 15.5% for males) for the individuals within each sex.

## 4. Discussion

### 4.1. Ecological Role in an Urban Landscape

Our study provided high taxonomic resolution of prey consumed by cane toads in southwest Florida golf course communities, offering some insights into the ecological roles of predator and prey in the urban environment. Yellow-banded millipedes, billbug weevils, and big-headed ants dominate the diet of cane toads inhabiting these communities, and, much like the predator, these prey items are considered pests in areas of human habitation. Yellow-banded and long-flange millipedes, the latter a less important prey species and both non-native to Florida, are considered a nuisance when large numbers invade houses and other structures, but they are harmless to humans and ecologically beneficial as detritivores [58,59,60]. Similarly, big-headed ants and less important toad prey such as Florida carpenter and rover ants may nest next to the foundation or within the buildings themselves and can become a nuisance in these dwellings [61]. Nonetheless, ants also provide a variety of essential ecosystem functions including biological control of other insect pests [62,63]. Extensive use of mulch with ornamental plantings in residential communities provides habitat for millipedes, ants, and earwigs [64,65,66]. Regarding the latter, the shore earwig was also a cane toad prey item of lesser importance and considered both an urban pest and a potential biocontrol agent against agricultural insect pests [67,68]. Hunting billbugs and, to a lesser degree, uneven billbugs are damaging pests in urban turfgrass systems [69,70,71], posing a potential aesthetic and economic impact on golf course communities owing to the expanse of grasses in lawns and fairways. Given that ants tend to be the dominant predators of insect eggs and larvae in turfgrass habitats [72] and some species of big-headed ant feed on weevil larvae [73,74], it would be interesting to determine whether big-headed ants and billbugs have a predator–prey relationship in golf course communities. Cane toads are feeding in the aforementioned anthropomorphic habitats as evidenced by observations of mulch and grass in the stomach samples and, just as many of these prey species may offer pest regulation services, so may cane toads by consuming these terrestrial arthropods perceived as urban pests. However, there are as of yet no quantitative data to indicate cane toads actually suppress pest populations [75] and experimental manipulations would be needed to demonstrate their predatory efficacy as has been performed with other purported biocontrol agents [67,73,74]. Nonetheless, a complex food web becomes apparent when using higher prey resolution [76] with ecologic functions such as pest predation by multiple taxa at multiple trophic levels within the anthropogenically altered ecosystem.

### 4.2. Taxonomic Resolution of Prey

The consumption of beetles, millipedes, and ants by cane toads in southwest Florida urban areas was similar to that of the prey reported in other diet studies for the species; however, the ordinal level of prey identification traditionally used in most studies, including those in Florida, precludes fine-scale comparisons and contrasts with the higher prey resolution of the current study. Furthermore, the limitations of lower taxonomic resolution in these other studies may complicate dietary interpretations and inferences regarding how cane toads utilize their dietary resources [35,76]. For example, Meshaka and Powell [33] demonstrated niche overlap between the cane toad and the native southern toad in central Florida using ordinal prey identification, and suggested the possibility of competition, provided prey resources were limited, owing to their broad overlapping diets of coleopterans (beetles) and hymenopterans (ants). In an earlier study, Meshaka and Mayer [77] identified the southern toad as an ant and beetle specialist in southern Florida when food items were identified “most often” to the familial level. Granted, other factors may have contributed to the difference between these studies, but it has been demonstrated that the level of taxonomic resolution can influence assumptions of prey utilization [34]. Dietary overlap appears substantially higher when the taxonomic resolution of prey is low and a decrease in overlap can be revealed with higher prey resolution [34,78,79,80].

Similarly, inferences regarding feeding strategies and food specialization can be influenced by the level of taxonomic resolution [34]. Most researchers have classified cane toads as indiscriminate, opportunistic, and/or generalist feeders based on ordinal or, at best, familial prey identifications and assumed that their diet is representative of the prey available in the respective study areas. Many of these studies have nonetheless identified beetles and ants as the dominant food items and a few have even suggested a preference for these prey items [80,81]. However, establishing this species’ foraging strategy and testing for food preferences requires the quantification of prey availability and efforts such as these are lacking in cane toad diet studies [10]. Strüssmann et al. [82] and Kidera et al. [83] concluded that cane toads consumed ants to a greater extent than their availability while Heise-Pavlov and Longway [84] also identified ants as primary prey, but sampling techniques for prey availability in the latter study yielded different compositions and thus different conclusions regarding food preferences. Nevertheless, these studies only offered qualitative contrasts of prey quantities consumed by toads with those available in their respective study areas. Further quantitative studies are needed using preference (i.e., electivity) indices or performing statistical tests of prey use relative to their availability in a given area with higher taxonomic identification of prey. However, not all prey necessarily need to be identified at higher taxonomic levels. A mix of taxonomic levels, as used in the current study, may perform better than those aggregated to genus or family level [38] provided any possible ambiguity among taxonomic levels has been addressed [55].

In addition to prey resolution, the choice of diet metrics may have influenced previous interpretations of cane toad diet. Most studies reported one or two diet metrics for prey taxa, such as frequency and/or number, and only a few measured the bulk (volume or mass) of food items (Appendix A). Only Parmalee [9] and Kidera et al. [83] reported the frequency, number, and volume of cane toad prey items, although the former had a very limited sample size (*n* = 5). Ideally, dietary investigations should employ at least one metric measuring amount (frequency and number), and one measuring the bulk of food items present in each sample [85]. Reporting multiple diet metrics can reveal potential bias in one or more of the measures that typically result from differences in the size of prey items [86]. As demonstrated in the current study, ants were numerous in cane toad samples but contributed very little to the prey volume, whereas considerably larger millipedes had an inverse relationship with these metrics. Our use of compound indices provided a balance for prey importance ratings with respect to prey size bias and identified important prey that would not have been elucidated had one or the other component metrics (number or volume) been used in analyses. 

### 4.3. Seasonal Differences in Diet

Despite individual variability and overlapping dietary compositions, cane toads exhibited significant seasonal differences in prey consumption in southwest Florida. Big-headed ants were the most important prey in both golf course communities during the early wet season and contributed the most to the differences for this season in both communities. Florida carpenter ants were also important wet-season prey in Community A. For regions with seasonal patterns of precipitation, cane toads ingested higher numbers of ants during the wet season in a coastal village of the Philippines [31] and with higher frequency in a severely fragmented tropical dry forest in Colombia [87]; however, neither study provided corresponding estimates of prey bulk in the diet. Ants were consumed at a higher proportional mass in arid coastal habitats during the Venezuelan wet season but were not accompanied by estimates of prey quantity [29]. Nonetheless, moisture is an important factor in regulating insect activities in the tropics and subtropics, and ants are typically more active during the wet season in these regions [88,89,90]. Illingworth [91] noted that cane toads ingested big-headed ants following rainy nights when the ants repaired their nest, and Zug and Zug [8] suggested toads may be attracted to the sound of ants repairing their nests (i.e., stridulation during excavation; [92]). Thus, behavioral patterns of both predator and prey could explain the prevalence of ants in the cane toad diet during the early wet season in southwest Florida. Future diet studies should compare the seasonal availability of ants with that ingested by toads to investigate whether there is any preference for ants during the wet season. Furthermore, our conclusions for seasonal prey consumption should be viewed cautiously given that the current study was limited to a single seasonal cycle (i.e., one wet season and one dry season). Multiannual sampling of cane toads is needed to determine whether the observed seasonal pattern of ant consumption persists and gauge the potential effects of environmental variability, particularly rainfall, on prey consumed by toads.

Prey compositions during the early dry season differed significantly from those in the early wet season for both golf course communities, but the importance of dry-season prey types also differed between communities. Hunting billbugs were the most important dry-season prey in Community A, while yellow-banded millipedes were more important in Community B, and both prey items contributed the most to the seasonal differences in the respective communities. The golf course communities in our study were relatively proximal to one another (approx. 500 m separated by barriers such as major roadways, canals, etc.) so location does not appear to be a factor; however, different human activities may be affecting arthropod ecology, or our perception as such, within each community [93]. Hunting billbugs are problematic turfgrass pests often managed with the application of synthetic insecticides [70,71], and insecticide use is known to impact arthropod ecosystems [94,95]. The comparably low dietary importance of billbugs in Community B during the dry season could have resulted from a more robust pest control regime within that golf course community prior to toad collection. However, pesticide use by the golf course communities and any such impacts on prey availability was beyond the scope of our study. Alternatively, cane toads were gathered by the pest control company based on ease of capture and their collection methods could have differed between communities. Favoring some areas over others could have resulted in unintentional bias with the collections. For example, the dry season stomach contents may reflect differences in the habitats sampled if toads were predominately collected on turfgrass in Community A and around mulched areas or near structures in Community B. A more rigorous sampling design is needed to control for possible differences among residential communities and their urban habitats with regard to the prey consumed by cane toads.

## 5. Conclusions

Yellow-banded millipedes, hunting billbug weevils, and bigheaded ants were the most important prey items for cane toads collected from golf course communities in southwest Florida. These arthropods are considered urban pests, but further studies are needed to determine whether cane toads provide any beneficial service with respect to the consumption of these nuisance taxa in the urban ecosystem. Our higher resolution for prey identification provided a better understanding of potential predator–prey relationships and their inferred use of habitats in the anthropogenically altered landscape. Furthermore, using multiple dietary metrics revealed size bias with the prey (i.e., numerous small ants versus fewer more voluminous millipedes), and applying compound indices provided a balance for interpretations of prey importance. There were significant seasonal differences in prey importance for cane toads, possibly related to the behavior of toads and the ants consumed during the wet season, and a locational difference in dry season prey importance that could have resulted from pest management practices in the golf course communities or unintentional bias during toad collections. We recognize the limited spatiotemporal scope of sampling effort in the current study and suggest a more rigorous, multiannual sampling design to address the potential effects of human activities and environmental variability on cane toad diet in urban ecosystems.

## Figures and Tables

**Figure 1 animals-13-02898-f001:**
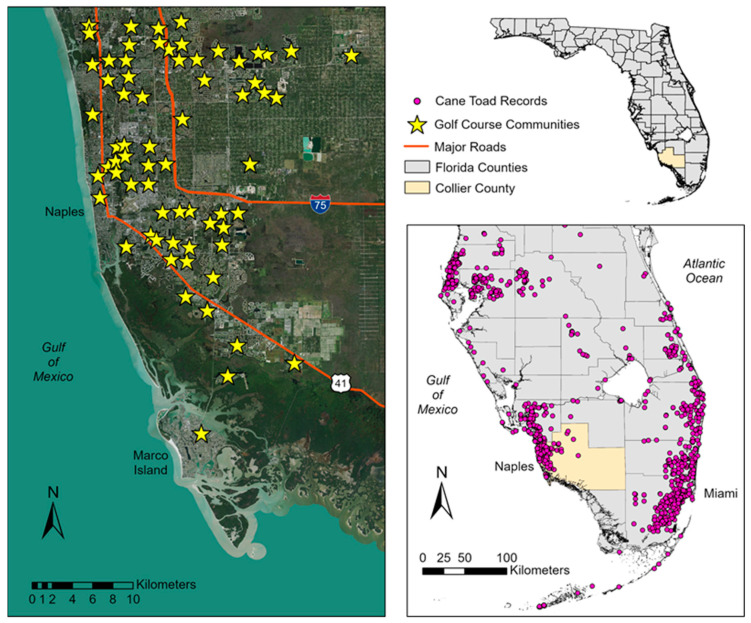
Maps showing the distribution records for cane toads in southern Florida and locations of golf course communities in Collier County, southwest Florida. Distribution records were obtained from the Florida Museum of Natural History Herpetology Collection [17], U.S. Geological Survey Nonindigenous Aquatic Species Database [18], and Early Detection and Distribution Mapping System (EDDMapS; [19]).

**Figure 2 animals-13-02898-f002:**
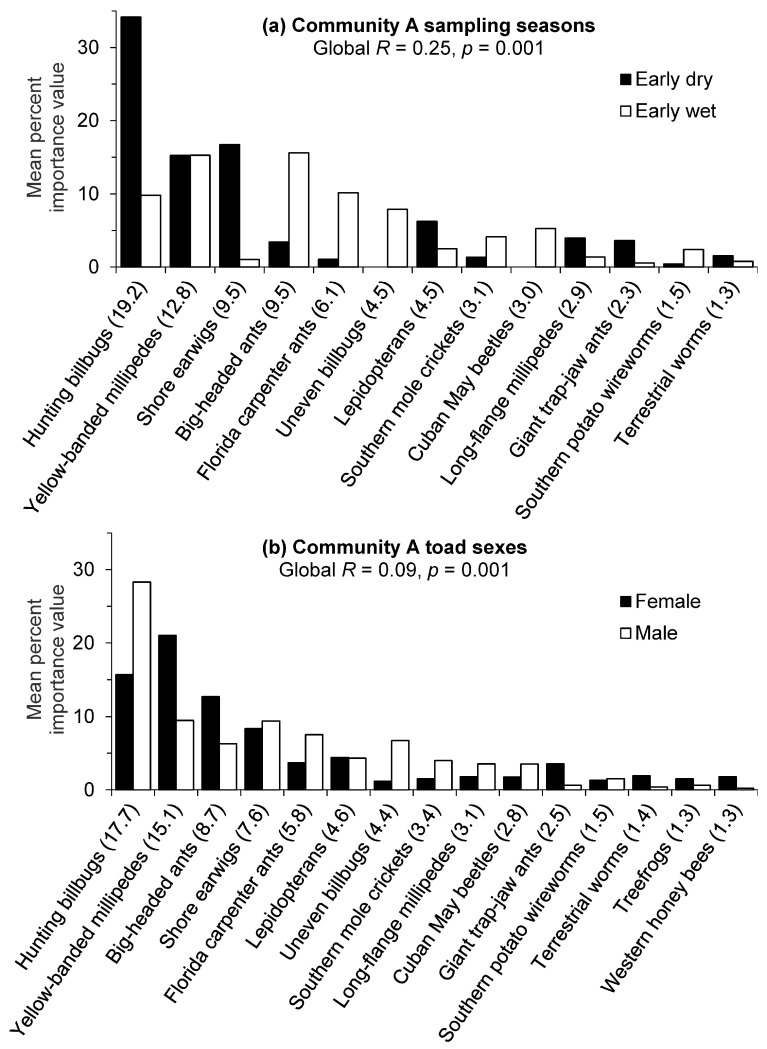
Cane toad prey contributions (80% cumulative) to the mean dissimilarity between (**a**) seasons and (**b**) sexes for golf course Community A in southwest Florida. Groups were contrasted by untransformed importance value percentages of prey items, and the contribution of each item to the total mean dissimilarity between groups is given in parentheses.

**Figure 3 animals-13-02898-f003:**
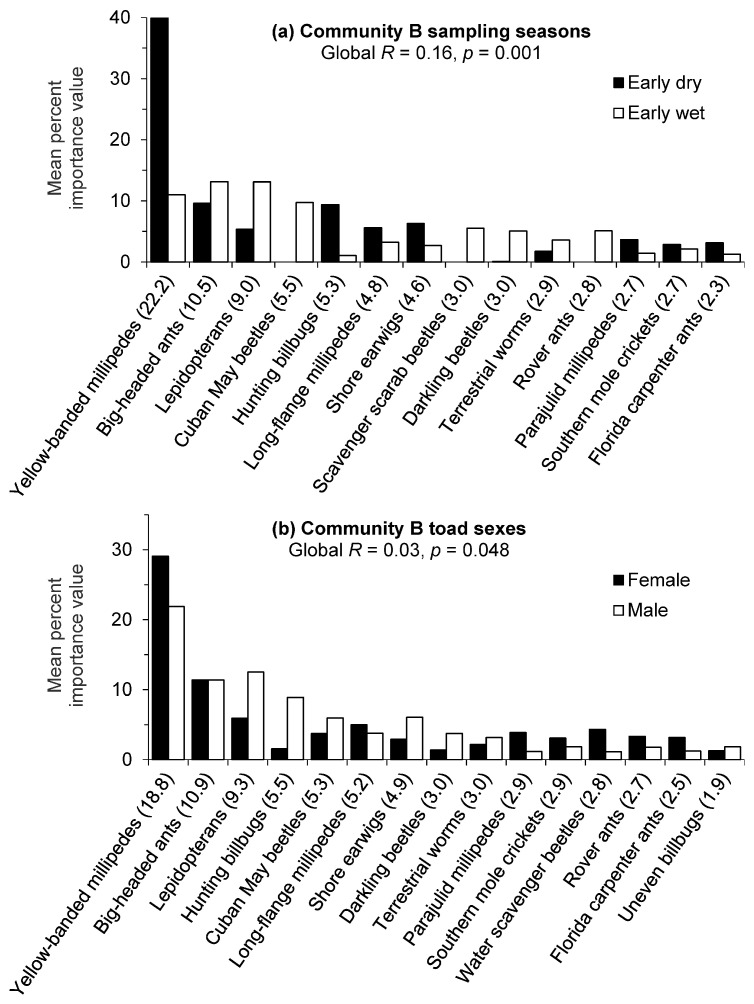
Cane toad prey contributions (80% cumulative) to the mean dissimilarity between (**a**) seasons and (**b**) sexes for golf course Community B in southwest Florida. Groups were contrasted by untransformed importance value percentages of prey items, and the contribution of each item to the total mean dissimilarity between groups is given in parentheses.

**Table 1 animals-13-02898-t001:** Seasonal summary of snout-urostyle lengths (mm) for cane toads collected from golf course communities in southwest Florida. Means are accompanied by ±one standard deviation and size ranges are given in parentheses. Different superscript letters indicate statistical significance.

Toad Sexes	Community A		Community B		F	*p*
	Early Dry	Early Wet	Early Dry	Early Wet		
Female	119.8 ± 16.1 ^b^	130.2 ± 12.5 ^a^	139.8 ± 26.3 ^a^	131.4 ± 12.3 ^a^	5.33	0.002
	(90–151)	(97–151)	(90–188)	(113–174)		
Male	116.8 ± 21.3 ^a^	119.7 ± 14.7 ^a^	122.8 ± 16.3 ^a^	121.0 ± 5.9 ^a^	0.57	0.64
	(73–197)	(62–142)	(95–150)	(111–131)		
t	0.62	2.98	3.01	4.19		
*p*	0.54	0.004	0.004	<0.001		

**Table 2 animals-13-02898-t002:** Seasonal percent number (%N), volume (%V), frequency of occurrence (%F), and index of relative importance (%IRI) for major food items of cane toads collected from golf course Community A in southwest Florida. Food items are ordered by IRI > 1% for at least one of the golf course communities.

Common Name (Scientific Name)	Early Dry				Early Wet			
	%N	%V	%F	%IRI	%N	%V	%F	%IRI
Yellow-banded millipedes (*Anadenobolus monilicornis*)	6.8	18.8	31.7	8.7	4.0	24.3	33.3	15.7
Hunting billbugs (*Sphenophorus venatus*)	56.8	35.4	68.3	67.8	5.5	6.7	36.7	7.4
Big-headed ants (*Pheidole*)	6.3	0.5	23.3	1.7	52.9	2.5	53.3	49.0
Butterflies/moths (Lepidoptera)	2.2	5.6	41.7	3.5	0.3	2.1	13.3	0.5
Shore earwigs (*Labidura riparia*)	11.7	12.3	61.7	15.9	0.4	0.7	13.3	0.3
Rover ants (*Brachymyrmex*)	0.1	<0.1	3.3	<0.1	2.9	<0.1	6.7	0.3
Uneven billbugs (*Sphenophorus inaequalis*)	0.0	0.0	0.0	0.0	7.1	7.9	31.7	7.9
Florida carpenter ants (*Camponotus floridanus*)	0.6	0.3	5.0	<0.1	11.0	2.0	33.3	7.2
Cuban May beetles (*Phyllophaga bruneri*)	0.0	0.0	0.0	0.0	1.2	6.3	21.7	2.7
Scavenger scarab beetles (*Hybosorus illigeri*)	0.0	0.0	0.0	0.0	0.7	2.2	11.7	0.6
Long-flange millipedes (*Asiomorpha coarctata*)	3.9	2.7	8.3	0.6	0.8	1.1	20.0	0.6
Southern potato wireworms (*Conoderus fallii*)	0.2	0.1	5.0	<0.1	1.5	0.9	43.3	1.7
Parajulid millipedes (Parajulidae)	0.4	0.1	5.0	<0.1	0.1	0.1	3.3	<0.1
Western honey bees (*Apis mellifera*)	<0.1	0.3	1.7	<0.1	4.3	7.8	5.0	1.0

**Table 3 animals-13-02898-t003:** Seasonal percent number (%N), volume (%V), frequency of occurrence (%F), and index of relative importance (%IRI) for major food items of cane toads collected from golf course Community B in southwest Florida. Food items are ordered by IRI > 1% for at least one of the golf course communities.

Common Name (Scientific Name)	Early Dry				Early Wet			
	%N	%V	%F	%IRI	%N	%V	%F	%IRI
Yellow-banded millipedes (*Anadenobolus monilicornis*)	43.2	65.3	56.1	79.5	0.9	18.3	31.6	10.2
Hunting billbugs (*Sphenophorus venatus*)	12.8	4.0	31.6	6.9	0.4	1.2	8.8	0.2
Big-headed ants (*Pheidole*)	11.2	1.4	29.8	4.9	55.2	6.7	47.4	49.3
Butterflies/moths (Lepidoptera)	1.9	2.9	10.5	0.7	6.7	17.6	47.4	19.4
Shore earwigs (*Labidura riparia*)	5.0	5.0	19.3	2.5	0.2	0.6	10.5	0.1
Rover ants (*Brachymyrmex*)	0.0	0.0	0.0	0.0	28.2	3.3	17.5	9.3
Uneven billbugs (*Sphenophorus inaequalis*)	0.6	0.3	5.3	0.1	0.4	1.5	12.3	0.4
Florida carpenter ants (*Camponotus floridanus*)	5.0	1.7	8.8	0.8	1.1	0.2	7.0	0.2
Cuban May beetles (*Phyllophaga bruneri*)	0.0	0.0	0.0	0.0	0.6	7.9	22.8	3.3
Scavenger scarab beetles (*Hybosorus illigeri*)	0.0	0.0	0.0	0.0	1.2	5.8	26.3	3.1
Long-flange millipedes (*Asiomorpha coarctata*)	8.1	3.2	15.8	2.3	0.3	1.7	14.0	0.5
Southern potato wireworms (*Conoderus fallii*)	0.1	0.1	1.8	<0.1	0.3	0.8	14.0	0.2
Parajulid millipedes (Parajulidae)	3.5	3.1	15.8	1.4	0.1	1.5	5.3	0.1
Western honey bees (*Apis mellifera*)	0.4	0.5	3.5	<0.1	<0.1	0.4	1.8	<0.1

## Data Availability

Data are available from authors upon request.

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
