# Peer review of "Faunal Diet of Adult Cane Toads, Rhinella marina, in the Urban Landscape of Southwest Florida"

_animals, 2023, doi:10.3390/ani13182898_

Round 1

Reviewer 1 Report

The manuscript entitled “Faunal Diet of Adult Cane Toads, Rhinella marina, in the Urban Landscape of Southwest Florida” by Melinda J. Schuman and colleagues, submitted to the Animals, presents results of a study on diet of cane toads inhabiting urbanized areas in southwest Florida. Such data are important to understand ecology of the species. I found the study interesting, and I think that the result should be published. I have several remarks which could help to improve the manuscript

Remarks
Lines 180-181 “The mean body length for male toads was smaller than that of females” – I believe it is true, however, result of statistical analyses is necessary here.

Table 1. There are ‘a’ and ‘b’ symbols. Thus, information that different letters indicate differences between groups is necessary.
There is information (in the Materials and method section) that all means are accompanied by ± one standard deviation. However, as in scientific papers, captions of figures and tables should be ‘self-explaining’, I recommend to add such information to the Table legend also.
Additionally, adding sample sizes of analysed groups is recommended.

In several places in the Results section there is phrase “and lower taxonomic resolution” – I am not sure if I precisely understand the phrase. Consider to change it, please.

Do you have any data on density or number of individuals of different groups of invertebrates/preys during the study? It would be interesting to show, if the toads mostly eat the most available invertebrates or present prey selection. The problem of ‘prey availability’ is discussed in Discussion section, however, data on number of individuals of different groups of invertebrates during the study would help to understand results of this study.

In the text there is “P” values, but on figures “p” values (small vs. capital letters, italics vs. normal types). Standardize the notation, please.

Please forgive me for my possible misunderstanding, but check the value percentages of prey items in the figure 3 – if the value 18.8 is correct?

As (lines 120-122) “Toads were collected by hand between 22:00 and 01:00 during November – December 2018 (early dry season) and June – July 2019 (early wet season).” data based on one ‘early dry season’ and one ‘early wet season’ are available only. Thus, it is necessary to be careful in drawing more general conclusions. For example, lines 410-413 “Big-headed ants were the most important prey in both golf course communities during the early wet season and contributed the most to the differences for this season in both communities.” It could suggest that in ‘wet’ seasons ants are the most important prey, however, data for one such ‘wet season’ are available, only. Consider to add some information in this subject, please.

Lines 145-147: the letters ‘?’ should be replaced by the symbols ‘×’.

Reviewer 2 Report

The paper titled “Faunal Diet of Adult Cane Toads, Rhinella marina, in the Urban Landscape of Southwest Florida” describes in a very comprehensive way the diet od the cane toad in anthropic environments like golf camps. The data are based on a very good sample size (240 individuals). Results are robust and the text is very well written and appreciable.

Only very few minor issues occur:

Line 39, speaking about “preadaptation” is quite questionable in evolutionary biology  (even if it is used by one of the two references cited) as it is difficult to demonstrated  that something adapted to a pressure that yet did not exist. It is a concept that may often also reflect a finalistic perspective not demonstrated by evidence. I would just remove the term “preadapted” from the sentence.

Line 115 it will be good to indicate the location of the two golf communities considered in the map reported in Figure 1.

Line 125, assessing  whether body condition affects diet can be add important information, as populations from different golf courses where grass management can differ may exhibit different activity periods and likely diverge in their peak of body condition. If also the wight is available it could be interesting to add ana analysis considering also BCI
